# Modified effect of active or passive smoking on the association between age and abdominal aortic calcification: a nationally representative cross-sectional study

Luyan Lv,[1] Shixian Wu,[2] Yungui Yang,[1] Xiongli Yue [ORCID] [1]

LL and SW contributed equally.

¹Department of Geriatrics, Qujing First People's Hospital, Qujing, Yunnan, China
²Ministry of Science and education, Qujing First People's Hospital, Qujing, Yunnan, China

**Correspondence to**
Dr Xiongli Yue;
yxl13529891626@163.com and Dr Yungui Yang;
1369687326@qq.com

## ABSTRACT

**Objective** The deleterious effects of smoking on atherosclerosis were well known; however, the interaction among ageing, smoking and atherosclerosis remains unclear. This study tested the hypothesis that the association between age and vascular calcification, a critical mark of atherosclerosis, was modified by smoking.

**Design** Cross-sectional study.

**Setting** A nationally representative sample, the National Health and Nutrition Examination Surveys 2013–2014.

**Participants** This study included 3140 adults aged 40–80 years with eligible data for abdominal aortic calcification (AAC). Active and passive smoking exposure was identified through self-reports and tobacco metabolites (serum cotinine and urinary 4-methylnitrosamino-3-pyridyl-1-butanol).

**Primary outcome measures** AAC score was determined using dual-energy X-ray absorptiometry (DXA) scans. OR was estimated using the logistic regression method to assess the association between age and the presence of severe or subclinical AAC stratified by smoking exposure. The survey-weighted Wald test was used to evaluate potential interactions.

**Results** AAC was positively associated with age in the general population. After adjustment for age, sex, race/ethnicity and other cardiovascular risk factors, age was significantly associated with the odds of severe AAC (OR for each 5-year increase in age: 1.66, 95% CI 1.48 to 1.87, p<0.001). As expected, the association between age and vascular calcification was especially stronger in smokers than in never smokers (p value for interaction ≤0.014). According to spline fitting, the progression of vascular calcification was significantly increased after 45 years in smokers compared with that after 60 years in never smokers. Quitting smoking may compromise the deleteriousness of the vascellum especially in younger adults. However, the difference in age-related calcification among never smokers with or without secondhand smoke exposure was minor, regardless of the definition by self-report, serum cotinine, or urinary 4-(methylnitrosamino)-1-(3-pyridyl)-1-butanol.

**Conclusions** Smoking significantly accelerated the progression of age-related subclinical atherosclerosis. Early smoking cessation should be encouraged among

## Strengths and limitations of this study

► This study involves a nationally representative sample of the adult population of the USA, suggesting an advantageous generalisation of the findings.
► Data were collected using standardised and validated protocols, which suggested favourable repeatability.
► This study evaluated the potential modification for age-related abdominal aortic calcification (AAC) by active smoking or secondhand smoke exposure, defined by self-reports or serum nicotine metabolites.
► This study was based on a cross-sectional design, which was less likely to conclude the causality.
► AAC measurements were only performed in participants aged 40 years or more. Whether smoking influences age-related vascular calcification in adults aged <40 years warrants further investigation.

young smokers. The effect of passive smoking exposure on arteriosclerosis should be assessed further.

## INTRODUCTION

Cardiovascular disease (CVD) has been the leading cause of death worldwide since the early 20th century, imposing a heavy burden on public health and healthcare systems.[1] Atherosclerosis has always been the primary pathological basis of CVD among the wide array of underlying cardiovascular lesions.[2] Early identification of populations at high risk of subclinical atherosclerosis is important to prevent the progression of CVD.[3] Abdominal aortic calcification (AAC) is considered a key hallmark of atherosclerosis and an independent predictor of subsequent cardiovascular morbidity and mortality.[1 4] Evidence from previous studies suggests that severe AAC is associated with a higher risk of major cardiovascular events, thus reflecting a poor cardiovascular health status.[5]

Epidemiological studies have demonstrated that vascular calcification or atherosclerosis is a common physiological process associated with ageing.[6 7] With the improvement in the primary and secondary preventions of cardiovascular events, the association between age and aortic calcification during the contemporary era remains unknown. Smoking is a well-established modifiable risk factor for atherosclerosis and CVD via several underlying pathways,[8] and it has been a critical environmental hazard for public health.[9] It is noteworthy that worldwide, approximately 11% of deaths related to CVD are attributed to smoking exposure.[10] Furthermore, second-hand smoke (SHS) exposure in non-smokers has also been associated with CVD and other health consequences.[11] Although several studies have demonstrated the associations between ageing or smoking and atherosclerotic events, the evidence on the interactive effect between ageing and active or passive smoking in aortic calcification is limited.[12] Clarifying this issue may provide new insights into the atherosclerotic process and identify the targeted population who could potentially benefit from smoking cessation education. In this context, this study tested the hypothesis that the association between age and AAC was modified by smoking exposure, including active and secondhand smoking, in a nationally representative sample of the middle-aged and elderly population.

## METHODS

### Study population

The data were extracted from the National Health and Nutrition Examination Surveys (NHANES) 2013–2014, an ongoing nationally representative-sampling, stratified-sampling, multistage probability-sampling survey designed by the Centers for Disease Control and Prevention of the USA. The methods and protocols have been described in previous studies.[13 14] There were 10 175 individuals in NHANES 2013–2014, and 3815 of them were adults aged 40 years of age and above. Pregnant women were excluded from the study (n=3). Dual-energy X-ray absorptiometry (DXA) scans were conducted for participants aged 40 years and above. Aortic calcification adjacent to the lumbar vertebrae L1–L4 was assessed. Participants without a scan (n=482) or with image invalidity (n=190) were excluded. All adults with eligible AAC scores were included in this study (n=3140).[15] Detailed information is available on the website http://wwwcdcgov/nchs/nhanes/irba98htm.

### AAC measurement

AAC could be easily assessed on lateral spine DXA scans.[16] The scores were calculated from a lateral scan of the lumbar spine (L1–L4 vertebrae) that was acquired using the Hologic Discovery model A densitometers (Hologic, Marlborough, Massachusetts, USA) and Apex software V.3.2.[1 16] The radiation exposure from DXA for the lateral spine scan was extremely low (<20 uSv). All scans were viewed using the Optasia SpinAnalyzer software.

The 24-point semiquantitative score of abdominal aortic calcification (AAC-24, Kauppila score) was used for the primary analysis, and the AAC-8 score (Schousboe score) was used for sensitivity analysis.[16] Both methods have been validated with favourable sensitivity and specificity to reflect the severity of AAC. AAC-24 was calculated according to the length of calcification at the posterior and anterior aortic walls contiguous to the L1–L4 lumbar vertebrae. Treating lines across the middle of the intervertebral spaces as segment boundaries, we found that the abdominal aorta was divided into eight segments. AAC was scored from 0 to 3 according to calcification length in the aortic wall of each segment (0 point: no calcification, 1 point: ≤1/3 arterial wall in each segment, 2 points: 1/3–2/3, 3 points: >2/3). AAC-8 is a simplified method derived from the AAC-24. AAC-8 in the anterior or posterior aortic walls in front of the L1–L4 was scored 0–4 (0 point: no calcification, 1 point: no more than the length of one vertebra, 2 points: no more than the length of two vertebrae, 3 points: no more than the length of three vertebrae and 4 points: more than the length of 3 vertebrae). Therefore, AAC-8 score is less influenced by small calcifications but need more skilled technologists compared with AAC-24. DXA examinations were performed by trained and certified radiology technologists. Detailed protocols have been presented in previous studies.[1 16]

### Smoking exposure and cigarette biomarker assessment

Smoking was defined as smoking at least 100 cigarettes during his or her life, ascertained through self-reports including both past and current cigarette smoking status. Tobacco metabolites, serum cotinine and urinary 4-(methylnitrosamino)-1-(3-pyridyl)-1-butanol (NNAL) were considered as biomarkers of passive smoking exposures in non-smokers.[17] Three definitions were used to identify individuals with SHS exposure, including self-report, urinary NNAL ≥0.001 ng/mL, or serum cotinine level of ≥0.015 ng/mL, as previously mentioned.[18] Serum cotinine and urinary NNAL were determined by high-performance liquid chromatography (HPLC)–tandem mass spectrometry, with atmospheric pressure chemical ionisation or electrospray ionisation, respectively. The biosample was pretreated with methyl-D3-cotinine or 4-(methylnitrosoamino)-1-(3-pyridyl)-1-butanol-1,2′,3′,4′,5′,6′-13 C6 as the internal standard. A C18 HPLC column was used to separate metabolites. The eluents from these injections were monitored using a mass spectrometer. The m/z 80 product ion from the m/z 177 quasi-molecular ion was used to quantify serum cotinine.

### Demographic, clinical and laboratory variables

Data concerning age, sex, race/ethnicity, smoking status, physical activity, alcohol consumption and history of chronic diseases were collected using standardised questionnaires during personal interviews.[13] Physical examinations were conducted at the Mobile Examination Centre (MEC) with standardised protocols. Peripheral blood

**Table 1** Characteristics of the study population in the National Health and Nutrition Examination Surveys 2013–2014

| Variable | Overall | Age ≤65 years | Age >65 years |
|---|---|---|---|
| Age (years) | 57.38 (56.75 to 58.01) | 51.89 (51.39 to 52.39) | 73.18 (72.72 to 73.65) |
| Male (%) | 48.08 | 49.3 | 44.56 |
| Race/ethnicity (%) | | | |
| Hispanic-Mexican | 6.98 | 8.03 | 3.98 |
| Other ethnicity | 11.89 | 13.0 | 8.69 |
| Non-Hispanic white | 71.01 | 68.2 | 79.1 |
| Non-Hispanic black | 10.12 | 10.78 | 8.23 |
| Smoking status (%) | | | |
| Never smoking | 53.98 | 54.38 | 52.84 |
| Former smoker | 28.15 | 24.27 | 39.33 |
| Current smoker | 17.87 | 21.35 | 7.831 |
| Physical activity (%) | | | |
| Inactive | 48.34 | 45.91 | 55.31 |
| Moderate | 32.75 | 31.67 | 35.86 |
| Vigorous | 18.91 | 22.41 | 8.83 |
| Alcohol intake (g) | 4.69 (4.01 to 5.36) | 5.31 (4.45 to 6.17) | 2.98 (2.39 to 3.56) |
| BMI (kg/m$^2$) | 28.54 (28.17 to 28.9) | 28.7 (28.25 to 29.15) | 28.07 (27.56 to 28.58) |
| Triglycerides (mmol/L) | 1.81 (1.73 to 1.89) | 1.84 (1.74 to 1.94) | 1.74 (1.63 to 1.85) |
| Total cholesterol (mmol/L) | 5.09 (5.04 to 5.13) | 5.18 (5.12 to 5.23) | 4.83 (4.70 to 4.96) |
| HDL-C (mmol/)L | 1.42 (1.40 to 1.43) | 1.41 (1.38 to 1.43) | 1.44 (1.39 to 1.48) |
| eGFR (mL/min/1.73 m²) | 84.28 (83.28 to 85.28) | 89.49 (88.26 to 90.72) | 69.47 (68.52 to 70.42) |
| SBP (mm Hg) | 125.42 (124.31 to 126.53) | 122.66 (121.43 to 123.89) | 133.28 (131.63 to 134.92) |
| DBP (mm Hg) | 70.75 (69.8 to 71.71) | 72.82 (72.03 to 73.61) | 64.86 (62.81 to 66.91) |
| Diabetes (%) | 14.32 | 11.39 | 22.74 |
| Lowering lipid (%) | 30.02 | 21.78 | 53.74 |
| ACEI/ARBs (%) | 26.87 | 20.23 | 45.97 |
| Beta blockers (%) | 14.63 | 8.89 | 31.14 |

Statistics of variables are presented as weighted proportion (%) or means (CI %).
ACEI, ACE inhibitor; ARB, angiotensin II receptor antagonist; BMI, body mass index; DBP, diastolic blood pressure; eGFR, estimated glomerular filtration rate; HDL-C, high-density lipoprotein cholesterol; SBP, systolic blood pressure.

was collected from the MEC and analysed in the central laboratory, according to standardised procedures. Race/ethnicity was categorised into four groups: non-Hispanic white, non-Hispanic black, Hispanic–Mexican and others. Physical activity was categorised as vigorous, moderate and less active as presented elsewhere.[19] Drinking was assessed via questionnaires for the preceding year. One drink was defined as 12 oz of beer, 4 oz of wine or 1 oz of liquor, each covering approximately 10 g of alcohol.[19] Body mass index (BMI) was calculated as weight divided by height squared (kg/m$^2$). Subjects' triglyceride (TG), total cholesterol (TC), high-density lipoprotein cholesterol (HDL-C) and serum creatinine levels were determined in morning fasting status. The estimated glomerular filtration rate (eGFR) was calculated using the Chronic Kidney Disease Epidemiology Collaboration.[13] Hypertension was defined as a self-reported diagnosis by a doctor, with systolic blood pressure of ≥140 mm Hg or a diastolic blood pressure

of ≥90 mm Hg. Diabetes was defined as receiving antihyperglycaemic therapy or glycated haemoglobin (HbA1c) levels of ≥6.5%.[19]

### Statistical analysis

Continuous and categorical variables are presented as means and percentages, weighted with primary sampling units, pseudo-strata and appropriate sampling weights to account for the complex sampling design. Crude and multivariable logistic or linear regression models were applied to assess the association between age and AAC scores. According to previous studies,[1 16] severe AAC was defined as an AAC-24 score of ≥6 points and subclinical AAC was defined as an AAC-24 score of ≥2 points. Logistic regression analysis was used to calculate ORs as primary analysis. Two adjusted models were used in this study. Model 1 was adjusted for sex and race/ethnicity. Model 2 was additionally adjusted for BMI, smoking

**Table 2** Association between age and severe or subclinical AAC in all participants or subgroups by smoking

| Models | Overall | | Never smoker | | Smoker | | P value for interaction |
|---|---|---|---|---|---|---|---|
| | OR (95% CI) | P value | OR (95% CI) | P value | OR (95% CI) | P value | |
| Severe AAC | | | | | | | |
| Crude | 1.84 (1.68 to 2.02) | <0.001 | 1.67 (1.50 to 1.85) | <0.001 | 2.03 (1.70 to 2.42) | <0.001 | 0.008 |
| Model 1 | 1.85 (1.69 to 2.02) | <0.001 | 1.67 (1.50 to 1.85) | <0.001 | 2.07 (1.74 to 2.46) | <0.001 | 0.008 |
| Model 2 | 1.66 (1.48 to 1.87) | <0.001 | 1.55 (1.32 to 1.82) | <0.001 | 1.78 (1.52 to 2.09) | <0.001 | 0.014 |
| Subclinical AAC | | | | | | | |
| Crude | 1.44 (1.32 to 1.56) | <0.001 | 1.29 (1.17 to 1.44) | <0.001 | 1.55 (1.43 to 1.68) | <0.001 | 0.002 |
| Model 1 | 1.44 (1.32 to 1.57) | <0.001 | 1.29 (1.16 to 1.43) | <0.001 | 1.55 (1.45 to 1.68) | <0.001 | 0.001 |
| Model 2 | 1.37 (1.25 to 1.50) | <0.001 | 1.26 (1.14 to 1.39) | <0.001 | 1.45 (1.31 to 1.61) | <0.001 | 0.011 |

The AAC-24 (Kauppila score) was calculated according to the length of calcification at the posterior and anterior aortic walls contiguous to the L1–L4 lumbar vertebrae. Treating lines across the middle of the intervertebral spaces as segment boundaries, we divided the abdominal aorta into eight segments. AAC was scored from 0 to 3 according to calcification length in the aortic wall of each segment (0 point: no calcification; one point: ≤1/3 arterial wall in each segment; 2 points: 1/3– 2/3; 3 points: >2/3). Severe AAC was defined as AAC-24 score ≥6 points, and subclinical AAC was defined as AAC-24 scores ≥2 points. The OR (95% CI) for the presence of severe or subclinical AAC was estimated using weighted logistic regression analysis. P value for interaction was used to assess the potential interaction between age and smoking for AAC using the survey-weighted Wald test. Model 1: adjusted for sex and race/ethnicity; model 2: additionally adjusted for body mass index, smoking status, alcohol consumption, physical activity, diabetes, hypertension, TC, TG, HDL-C, eGFR and decreasing lipid agents.

AAC-24, 24-point semiquantitative score of abdominal aortic calcification; AAC, abdominal aortic calcification; eGFR, estimated glomerular filtration rate; HDL-C, high-density lipoprotein cholesterol; TC, total cholesterol; TG, triglyceride.

status, alcohol consumption, physical activity, diabetes, hypertension, TC, HDL-C, eGFR, and decreasing lipid agents. The variance inflation factor of each variable was not more than 2.64, which did not suggest a multicollinearity of covariates. The survey-weighted Wald test was used to assess the potential interaction between age and smoking for AAC. The β coefficient (95% CI) was presented as the increased scores of AAC-24 or AAC-8 for each 5-year increase. Two-way fractional polynomial regression spline plots were used to visualise the relationship between age and vascular calcification among smokers or never smokers, and SHS or non-SHS. Statistical analysis was conducted with Stata V.SE15.0. All tests with a two-sided p value of less than 0.05 were considered statistically significant.

### Patient and public involvement

No patients were involved in the development of the research question or design of the study.

### RESULTS

Overall, there were 3140 adults aged ≥40 years with AAC scores in NHANES 2013–2014 (table 1). The weighted mean age was 57.4 years and 51.9% (n=1622) were women. According to self-report, 54.0% were never smokers and 46.0% were smokers, including 28.2% ever smokers and 17.8% current smokers. Compared with participants aged ≤65 years, older participants had higher proportions who were non-Hispanic white, quitting smoking, diabetes and cardiovascular medications, as well as less physical activity and lower renal function.

The association between age and AAC in all the participants, stratified by smoking, is shown in table 2 and online supplemental table 1. In univariable and sex-adjusted and race-adjusted logistic regression models, age was significantly associated with severe or subclinical ACC, with 85% and 44% increased odds for each 5-year increase in age (each p<0.001). After adjustment for sex, race, smoking, physical activity, alcohol intake, BMI, hypertension, diabetes, TC, TG, HDL-C, eGFR and decreasing-lipid agents, the association of age and severe or subclinical AAC remained significant with ORs of 1.66 (95% CI 1.48 to 1.87) and 1.37 (95% CI 1.25 to 1.50) for each 5-year increase in age (each p<0.001), respectively (table 2). In addition, AAC-24 and AAC-8 scores were positively associated with age (online supplemental table 1). The association between age and AAC-24 score remained significant (β coefficient for each 5-year increase in age: 0.43, 95% CI 0.33 to 0.52, p<0.001). Supplementary analysis for the association between smoking exposure and AAC was conducted (online supplemental tables 2–4). As expected, current smoking was significantly associated with AAC severity compared with never smokers. Treating current smokers as a reference, participants with smoking cessation of more than 10 years had lower AAC scores. However, the relationship between passive smoking and AAC was not observed.

Because smoking cigarettes are a strong risk factor for vascular injury, we assessed age-related aortic calcification in subgroups according to smoking status (table 2 and online supplemental table 1). As expected, the effect modification by smoking was observed (p value for interaction ≤0.014). The OR for each 5-year increase in age

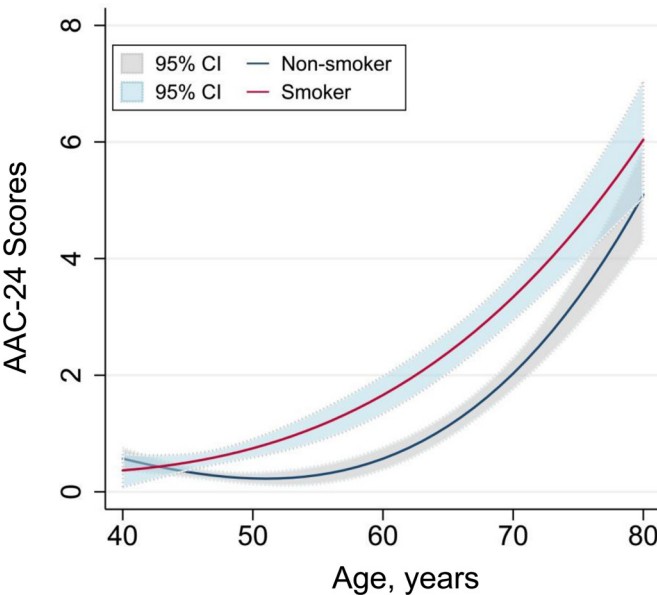

**Figure 1** Spline fitting of age-related AAC in never smokers and smokers. The spline curve shows the relationship between scores of AAC-24 and age in smokers and never smokers based on fractional polynomial regression. The solid lines represent point estimates, and dashed lines represent 95% CIs. AAC-24 was calculated according to the length of calcification at the posterior and anterior aortic walls contiguous to the L1–L4 lumbar vertebrae. Treating lines across the middle of the intervertebral spaces as segment boundaries, we divided the abdominal aorta into eight segments. AAC was scored from 0 to 3 according to calcification length in the aortic wall of each segment (0 point: no calcification, 1 point: ≤1/3 arterial wall in each segment, 2 points: 1/3– 2/3; 3 points: >2/3). Never smokers had a flat trend of age-related calcification before the age of 65, whereas the calcification was significantly advanced after the age of 45 in subjects with smoking. AAC, abdominal aortic calcification; AAC-24, 24-point semiquantitative score of abdominal aortic calcification.

for severe AAC was 1.67 (95% CI 1.50 to 1.85) among non-smoking participants vs 2.03 (95% CI 1.70 to 2.42) among smoking participants. This suggests that the ever and current smoking may accelerate age-related AAC by 30% compared with never smoking status. Repeated analysis was performed using binary subclinical AAC and continuous AAC-24 and AAC-8 scores, and the correlations remained unchanged. According to spline fittings (figure 1), never smokers had a flat trend of age-related calcification before 65 years, whereas the calcification was significantly advanced after 45 years in smoking participants. Furthermore, compared with current smokers, participants who quit smoking had lower scores for age-related calcification, especially for those aged 50–70 years (figure 2). However, among never smokers, SHS exposure did not significantly affect age-related AAC (figure 3).

## DISCUSSION

In this nationally representative sample from the general population of the USA in 2013–2014, the association

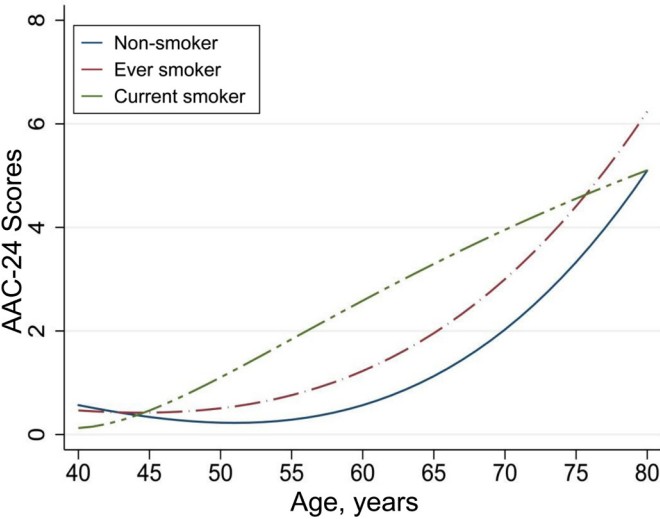

**Figure 2** Visualisation of age-related AAC in smokers who have quit smoking or not. The spline curve shows the relationship between scores of AAC-24 and age in smokers who quit smoking or not based on fractional polynomial regression. The solid lines represent point estimates. AAC, abdominal aortic calcification; AAC-24, 24-point semiquantitative score of abdominal aortic calcification.

between AAC and age differed, depending on the smoking status after adjustment for various confounders. Age-related calcification was flat before 60 years old, whereas it was driven by smoking after 45 years old. Compared with persistent smoking, quitting smoking may mitigate the process of age-induced calcification, especially for adults aged 45–70 years. The deleterious effect of secondhand smoking on vascular calcification in adults was not observed in this study. Our study is of particular interest because our findings highlight the benefits of never smoking and early quit smoking in the process of age-related vascular calcification or atherosclerosis in the contemporary population.

Vascular calcification was considered to be a degenerative and end-stage consequence of the ageing process.[6] Arterial calcification was detected in approximately one-fourth of the populations aged 50 years. This proportion subsequently increased to beyond 60% in the population aged more than 75 years.[7] Our results also consistently noted that AAC scores slowly increased in middle-aged participants, and robustly increased in the elderly participants aged ≥65 years.

Vascular calcification has been linked with vascular injury and subclinical atherosclerotic disease.[16 20] Vascular wall calcification is a common byproduct of atherosclerotic plaques and was thus considered as a potential marker of subclinical atherosclerosis as well as a simple tool for cardiovascular risk stratification.[16] Although previous reports have highlighted the implications of AAC as a biomarker of cardiovascular morbidity and mortality in adults,[21–23] AAC has received less attention as a modifiable marker in preventing atherosclerosis and cardiovascular risk.[24] Our findings suggest that quitting smoking is a simple and effective means to mitigate the process

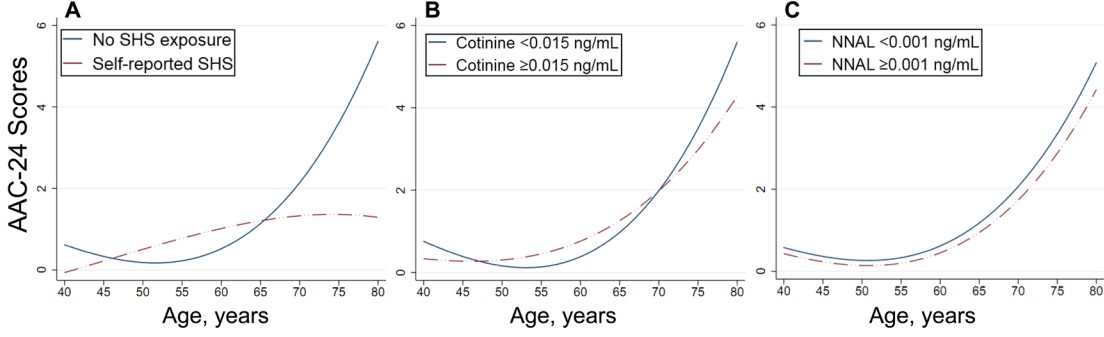

**Figure 3** Age-related vascular calcification in never smoker with or without SHS exposure. Secondhand smoking exposure was defined by self-reports (A), serum cotinine (B) and urinary NNAL levels (C), respectively. The spline curve shows the relationship between AAC-24 scores and age in never smokers with or without SHS exposure based on fractional polynomial regression. AAC-24, 24-point semiquantitative score of abdominal aortic calcification; NNAL, 4-(methylnitrosamino)-1-(3-pyridyl)-1-butanol; SHS, secondhand smoke.

of arterial calcification and atherosclerosis in the general population, especially in young adults. Lifestyle changes seem to be promising, low-cost and easy to spread.

Hisamatsu *et al* conducted an interesting study and demonstrated a positive association between smoking and subclinical atherosclerosis that was comprehensively assessed based on coronary artery calcification, carotid intima–media thickness and plaque, aortic artery calcification (from the aortic arch to iliac bifurcation) and ankle–brachial index.[25] The relationship attenuated over the time of smoking cessation. Moreover, Pham *et al* supplemented the analysis for the relationship between smoking habits and the progression of coronary and aortic artery calcification in a follow-up study.[26] These findings supported that smoking is a strong risk factor for coronary calcification and atherosclerosis. Jung and colleagues observed the positive association between AAC and long-term smoking exposure in 218 men aged 40–81 years. Even a low dosage of smoking increased the risk of AAC.[27] Furthermore, Hirooka and colleagues analysed 313 Japanese and 302 Korean men aged 40–49 years and concluded that smoking with higher pack-years was significantly associated with artery calcium, suggesting a dose–response pattern between cigarette use and progression of atherosclerosis.[28] A small sample size and selection bias may limit the extrapolation. Our analysis also supported the increased risk of severe or subclinical angiosteosis associated with smoking. Notably, smoking cessation for more than 10 years could partly attenuate the harm of smoke exposure, which was consistent with the findings of previous reports.[25 26 29] Our results further emphasised that the association between age and ACC was modified by smoking and that the detrimental effects of ageing and smoking were not simply superposed.

Based on NHANES data, a recent study noted that serum cotinine levels, a biomarker of smoking, were associated with the risk of severe AAC in 2840 adults.[16] However, circulating cotinine only reflects short-term exposure to smoking, and the interaction between smoking and age on the process of vascular calcification remains unknown. Our results highlighted that the

process of age-related calcification was slow before 60 years of age. In contrast, smoking, especially persistent smoking, significantly accelerated this process since 40 years of age. Quitting smoking at an early stage may at least partially moderate the speed for adults aged ≤70 years. To the best of our knowledge, our study is the first study that specifically investigates the interaction between age and smoking on the severity of aortic calcification in humans. Notably, previous studies have established the association between smoking and vascular calcification, suggesting the potential benefits of quitting smoking to prevent atherosclerosis.[8 16 30 31] However, the complex interaction with the ageing process needs to be clarified, which may provide new insights into the optimal time to prevent atherosclerosis by tobacco control.[30] Our results suggest small benefits of quit smoking in the elderly aged >70 years, which highlights the importance of early action to reduce the burden of vascular calcification.

SHS exposure has been proven to be associated with CVD and adverse outcomes.[11] Approximately 85% of SHS was derived from side-stream smoke that seemed potentially more harmful due to non-filtering.[12] As another purpose, we investigated the relationship between AAC and SHS based on self-reports, and serum cotinine or urine NNAL in never smokers. However, the neutral link between arterial calcification and passive smoking that we noted might be not the ultimate conclusion. The questionnaires of secondhand smoking were only designed for the preceding 7 days, although we used the definition of potentially persistent SHS exposure in indoor settings at job or home. Serum cotinine and urinary NNAL, as more accurate biomarkers of smoking exposure than self-reports, have been validated to estimate the SHS exposure in the past 2–4 days and 6–12 weeks, respectively.[18] The relationship was also insignificant in stratifications by serum cotinine or urinary NNAL. Both biomarkers only reflected exposures over the past several days or weeks, which were insufficient to reflect long-term exposure. The progression of vascular calcification is a chronic process, even beyond the years. Therefore, further studies may need multiple repeated measurements or questionnaires

over the years to assess the effect of SHS on age-related calcification.

## Limitations and strengths

Our study has some strengths that support favourable repeatability, including a nationally representative sample of the adult population, standardised and validated protocols, and full adjustment for a large number of potential confounders. However, our conclusions should be explained in consideration of the following limitations. First, this study was based on a cross-sectional design, which was less likely to conclude the causality. More convincing evidence could be obtained from a longitudinal study with multiple measurements. Second, the analysis of the relationship between SHS and AAC was not the ultimate seal. The duration of SHS exposure should be considered in future studies. Third, AAC measurement was performed only in the population aged 40 years or above. Whether smoking influences age-related vascular calcification in adults aged <40 years old warrants investigation. Fourth, residual confounding factors could not be eliminated, although we strictly adjusted for 16 covariates in the regression model.

## CONCLUSIONS

Our results indicate that smoking significantly modifies the association between age and AAC severity in adults aged 40 years or above. Age-related vascular calcification is accelerated by smoking, which is partly reduced by smoking cessation. However, whether SHS affects this process requires further investigation.

**Acknowledgements** The authors thank all participants and staff in National Health and Nutrition Examination Surveys for their great contributions.

**Contributors** YY and XY contributed to the conceptualisation and developed the protocols. LL and SW organised and analysed all data. XY was responsible for the project administration. All authors wrote and reviewed the manuscript, and approved the final manuscript. YY and XY accepted full responsibility for the finished work and the conduct of the study, had access to the data, and controlled the decision to publish.

**Funding** The authors have not declared a specific grant for this research from any funding agency in the public, commercial or not-for-profit sectors.

**Competing interests** None declared.

**Patient consent for publication** Consent obtained directly from patient(s).

**Ethics approval** The protocols and procedures of the National Health and Nutrition Examination Surveys(NHANES) study were agreed by the Research Ethics Review Board of the Centers for Disease Control and Prevention of the USA (protocol number. 2011-17). This study was conducted in accordance with the Declaration of Helsinki. The protocols of the NHANES were approved by the Ethics Review Board of the National Center for Health Statistics. All participants provided informed consent.

**Provenance and peer review** Not commissioned; externally peer reviewed.

**Data availability statement** All data came from the NHANES study (http://www. cdc.gov/nchs/nhanes). Data are available on reasonable request from the CDC of US or corresponding authors.

**ORCID iD**

Xiongli Yue http://orcid.org/0000-0002-2937-6824

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
