## [Reviewer comments · BMJ Open]

ARTICLE DETAILS

TITLE (PROVISIONAL)	Modified effect of active or passive smoking on the association between age and abdominal aortic calcification: a nationally representative cross-sectional study
AUTHORS	Lv, Luyan; Wu, Shixian; Yang, Yungui; yue, xiongli

VERSION 1 – REVIEW

REVIEWER	Hisamatsu, Takahashi Okayama University Graduate School of Medicine Dentistry and Pharmaceutical Sciences, Department of Public Health
REVIEW RETURNED	02-Feb-2021

GENERAL COMMENTS	The authors present a cross-sectional observation from the NHANES 2013-2014 in 3140 US individuals aged 40-80 years to test the hypothesis that the association of age with abdominal aortic calcium burden is modified by the smoking exposure. The strength of the present study is the use of nationally representative sample from the US general population. Data including smoking exposure and abdominal aortic calcium score were based on standardized and validated protocol. However, there are some concerns in this paper. Major 1. Introduction. As the authors mentioned (page 6, 2nd paragraph), vascular calcification or atherosclerosis is a common aging-related process. Smoking is also well-established risk factor for vascular calcification or atherosclerosis. It is obvious that smoking exposure affects the association between age and aortic calcium burden. Therefore, the scientific premise of the present study seems to be unclear for me.2. Study population. How many US adults participated to the NHANES in 2013-2014? How many of them underwent DXA scans to measure abdominal aortic calcium score? What is the participation rate? Did the authors consider the possibility of selection bias?3. Outcome, abdominal aortic calcium score. The definitions of AAC-24 and AAC-8 scores remain unclear. Please describe them more specifically in the Methods section.4. Statins promote vascular atheroma calcification which ultimately results in plaque stabilization and less likelihood for progression to unstable plaques. I suggest the authors take statin use into account in the analysis.5. Statistical analysis. In general, aortic calcium score is likely to show skewed distribution, but not normal distribution. Did the
---

authors confirm its distribution? I do not think it is appropriate to analyze the data using the linear regression.

6. Too much covariates in a multivariable model may cause the problem of overfitting. Did the authors examine the multicollinearity of covariates when performing multivariable analysis? For example, hypertension was defined as systolic/diastolic BP of $\geq 140/90$ mmHg or antihypertensive medication use. However, hypertension, ACEI/ARBs use, and beta blocker use were simultaneously included in the multivariable model.

7. Results. It will be important to present the association of smoking status (current, former, and never smoking) with abdominal aortic calcium burden as one of the results.

8. Discussion, Page 15, 2nd paragraph, "the data on smoking and abdominal arterial calcification were limited". However, some prior studies provided evidence in this area: Hisamatsu et al. *J Am Heart Assoc.* 2016 Aug 29;5(9):e003738; and Pham et al. *Int J Cardiol.* 2020 Sep 1;314:89-94.

9. The authors conclude that age-related vascular calcification is partly reduced by quit smoking. To obtain more solid scientific evidence, the authors should examine the association of age with abdominal aortic calcium burden based on current smoking, former smoking groups according to years since quit smoking vs. never smoking.

Minor

1. Page 9, line 3. The sentence "The detailed protocols have been presented in previous studies" seems to need the appropriate references.
2. I suggest the results based on AAC-8 be also provided in the supplement.
3. Page 16, line 6, "suggesting the potential benefits of quit smoking to prevent atherosclerosis". Please consider prior 3 papers which reports the association of time since quitting in former smokers with atherosclerosis burden (McEvoy et al. *Arterioscler Thromb Vasc Biol.* 2015 Apr;35(4):1002-10; Hisamatsu et al. *J Am Heart Assoc.* 2016 Aug 29;5(9):e003738; Kianoush et al. *J Am Heart Assoc.* 2017 Jun 24;6(6):e005088.)
4. Please change "non smoker" to "never smoker".
5. Table 1. Why did the authors divided participants into 2 groups based on 65 years old? Please describe the reasons in the statistical analysis of the Methods section.
6. Figure 1-3, legends. Please explain the definition of AAC-24.

VERSION 1 – AUTHOR RESPONSE

Comments to the Author:

The authors present a cross-sectional observation from the NHANES 2013-2014 in 3140 US individuals aged 40-80 years to test the hypothesis that the association of age with abdominal aortic calcium burden is modified by the smoking exposure. The strength of the present study is the use of nationally representative sample from the US general population. Data including smoking exposure and abdominal aortic calcium score were based on standardized and validated protocol. However, there are some concerns in this paper.

Response: We would like to thank reviewer for the careful reading and helpful comments. Each comment was presented with a reply that we do our best. If the answer is inaccurate or insufficient, Please point out. Thanks for your patient review.

Major

1. Introduction. As the authors mentioned (page 6, 2nd paragraph), vascular calcification or atherosclerosis is a common aging-related process. Smoking is also well-established risk factor for vascular calcification or atherosclerosis. It is obvious that smoking exposure affects the association between age and aortic calcium burden. Therefore, the scientific premise of the present study seems to be unclear for me.

Response: We speculate that reviewers refers to confounding rather than interaction. We agree with the reviewer that a variety of cardiovascular risk factors such as age, sex, smoking, and metabolic syndrome are associated with vascular calcification. Each risk factor independently contributes to the outcome and results in cumulative effects. Therefore, multivariable analysis is used to adjust for smoking and other risk factors (potential confounders) and observe the independent correlation between age and vascular calcification. However, that does not mean smoking and age have an interaction on calcification, the amplification effect produced by the combination of aging and smoking. Thomas and Edward recently reviewed the difference between confounding and interaction and the application (Thomas R Vetter, Edward J Mascha. Bias, Confounding, and Interaction: Lions and Tigers, and Bears. *Anesth Analg*. 2017;125(3):1042-1048). We drew a graph of the difference between the two as followed. To avoid ambiguity, we have reorganized the introduction. Thank you very much. (P6/7)

2. Study population. How many US adults participated to the NHANES in 2013-2014? How many of them underwent DXA scans to measure abdominal aortic calcium score? What is the participation rate? Did the authors consider the possibility of selection bias?

Response: Thanks for your comments. We have added a detailed description in Methods. There were 10,175 individuals in NHANES 2013-2014 and 3,815 adults 40 years of age and older.. Pregnant females were excluded (n=3). dual-energy X-ray absorptiometry (DXA) scans were conducted in participants aged 40 years and older. Aortic calcification for vertebrae lateral spine L1–L4 was assessed. Excluding participants without scan (n=482) or image invalidity (n=190). All those with eligible AAC scores were included in this study (n = 3140).

NHANES study is an ongoing nationally representative, stratified, multistage probability-sampling survey designed by the Center for Disease Control and Prevention of the United States. NHANES specially provided several subsample weights to minimize the selection. For NHANES analysis, using the correct sample weight depends on the variables used. A good rule of thumb is to use the "lowest common denominator", where the variable of interest collected on the smallest number of respondents is the "lowest common denominator". The sample weight applied to that variable is the appropriate weight for that particular analysis. (<https://wwwn.cdc.gov/nchs/nhanes/tutorials/module3.aspx>) (P8)

3. Outcome, abdominal aortic calcium score. The definitions of AAC-24 and AAC-8 scores remain unclear. Please describe them more specifically in the Methods section.

Response: Thanks for your suggestions. We added the definitions of AAC-24 and AAC-8 scores. Abdominal aortic calcification could be easily assessed on lateral spine scans obtained by DXA with some advantages including inexpensive, ease, rapid and safe. AAC-24 was calculated according to the length of calcification at the posterior and anterior aortic walls contiguous to lumbar spine L1-L4. Treating lines across the middle of intervertebral spaces as segment boundaries, abdominal aorta was divided into 8 segments. AAC was scored from 0 to 3 according to calcification length in aortic wall of each segment (0 point: no calcification; 1 point: $\leq 1/3$; 2 points: $1/3-2/3$; 3 points: $>2/3$). The AAC-8 was a simplified method derived from AAC-24. Aortic calcification in anterior or posterior aortic walls in front of L1-L4 was scored 0 to 4, respectively (0 point: no calcification; 1 point: no more than the length of one vertebrae; 2

points: no more than the length of 2 vertebra; 3 points: no more than the length of 3 vertebra; 4 points: more than the length of three vertebra). Therefore, AAC-8 score was less influenced by small calcification but needed more skillful technologists than AAC-24 (Pawel Szulc. Bone. 2016 Mar;84:25-37) . (P9)

4. Statins promote vascular atheroma calcification which ultimately results in plaque stabilization and less likelihood for progression to unstable plaques. I suggest the authors take statin use into account in the analysis.

Response: Thanks for your comments. we agreed with the reviewer that statin treatment improved lipid metabolism and plaque stabilization. In our manuscript, multivariable regression analysis included lowering lipid agents, of which statins accounted for 94%. Furthermore, the variable “lowering lipid agents” was replaced by “Statins treatment” and the results was consistent.

5. Statistical analysis. In general, aortic calcium score is likely to show skewed distribution, but not normal distribution. Did the authors confirm its distribution? I do not think it is appropriate to analyze the data using the linear regression.

Response: Thank the reviewer to the professional comments. We transformed AAC-24 scores as binary data. According to previous studies, severe AAC was defined as AAC-24 ≥ 6 points and subclinical AAC was defined as AAC-24 ≥ 2 points. The distribution of aortic calcium score was skewed. But the residuals of linear regression model was close to a normal distribution which was emphasized as a necessary condition in generalized linear model (including linear, logistics and Cox regression). The trend in both models was consistent. We agree with the rigorous manner of the reviewer. Logistics regression to calculate odds ratios was shown in Table 1 as primary analysis. And the linear regression results was presented in supplementary document.

6. Too much covariates in a multivariable model may cause the problem of overfitting. Did the authors examine the multicollinearity of covariates when performing multivariable analysis? For example, hypertension was defined as systolic/diastolic BP of $\geq 140/90$ mmHg or antihypertensive medication use. However, hypertension, ACEI/ARBs use, and beta blocker use were simultaneously included in the multivariable model.

Response: Thanks for your comments. We agreed with the reviewer that over-fitting should be considered. More than two third users of ACEI/ARBs or β -blocker had hypertension. We removed the above two variables and multivariable models were assessed by variance inflation factor (VIF). VIF of each variable was not more than 2.64 that did not suggest a multicollinearity of covariates. We supplemented necessary descriptions in methods.(P12)

7. Results. It will be important to present the association of smoking status (current, former, and never smoking) with abdominal aortic calcium burden as one of the results.

Response: we supplemented the analysis for the correlation between smoking status and ACC (Table S2).

8. Discussion, Page 15, 2nd paragraph, “the data on smoking and abdominal arterial calcification were limited”. However, some prior studies provided evidence in this area: Hisamatsu et al. J Am Heart Assoc. 2016 Aug 29;5(9):e003738; and Pham et al. Int J Cardiol. 2020 Sep 1;314:89-94.

Response: Hisamatsu et al. conducted an interesting study to demonstrate a positive association between smoking and subclinical atherosclerosis assessed by coronary artery calcification, carotid intima-media thickness and carotid plaque, aortic artery calcification (from aortic arch to iliac bifurcation), and ankle-brachial index comprehensively. The relationship attenuated with the time of smoking cessation. Moreover, Pham et al. further supplemented the analysis for the relationship between smoking habits and

progression of coronary and aortic artery calcification in a follow-up study. Although abdominal arterial calcification was not specially measured, risk factors and process of arterial calcification in aortaventralis were comparable to that in coronary and other aortic artery. We added analysis about arterial calcification and smoking habits in discussion. Our results emphasized that the association between age and ACC was modified by smoking, and the detrimental effects of ageing and smoking were not simply superposed. In addition, we initially explored whether second-hand smoking affected age-related arterial calcification. Thanks for the reviewer's suggestion. (P16)

9. The authors conclude that age-related vascular calcification is partly reduced by quit smoking. To obtain more solid scientific evidence, the authors should examine the association of age with abdominal aortic calcium burden based on current smoking, former smoking groups according to years since quit smoking vs. never smoking.

Response: We supplemented the sensitivity analysis. Smokers were categorized into 3 groups: quitting for more than 10 years, less than 10 years and current smoking. Individuals who quit smoking had a lower risk of AAC compared with current smokers (Table S3). Thank you for the suggestion.

Minor

1. Page 9, line 3. The sentence “The detailed protocols have been presented in previous studies” seems to need the appropriate references.

Response: Necessary references was supplemented. Thanks for your kindly reminder.

2. I suggest the results based on AAC-8 be also provided in the supplement.

Response: Thanks for your suggestions. Binary AAC-24 was investigated as the primary analysis. the analysis of AAC-8 was shown in the supplementary Table S1-S2.

3. Page 16, line 6, “suggesting the potential benefits of quit smoking to prevent atherosclerosis”. Please consider prior 3 papers which reports the association of time since quitting in former smokers with atherosclerosis burden (McEvoy et al. *Arterioscler Thromb Vasc Biol.* 2015 Apr;35(4):1002-10; Hisamatsu et al. *J Am Heart Assoc.* 2016 Aug 29;5(9):e003738; Kianoush et al. *J Am Heart Assoc.* 2017 Jun 24;6(6):e005088.)

Response: Thanks for your suggestion. We have added a discussion about time to quit smoking and atherosclerosis. (P17 L2/7)

4. Please change “non smoker” to “never smoker”.

Response: We have carefully checked and revised the descriptions. Thanks very much.

5. Table 1. Why did the authors divided participants into 2 groups based on 65 years old? Please describe the reasons in the statistical analysis of the Methods section.

Response: In this study, we intended to assess whether the association between and AAC was modified by smoking. Age and AAC was the primary exposure and outcome, respectively. Grouping by exposure factors can initially assess which variables are related to aging.

6. Figure 1-3, legends. Please explain the definition of AAC-24.

Response: The definition of AAC-24 was supplemented in Methods section and figure legends (P8/9).

REVIEWER	Hisamatsu, Takahashi Okayama University Graduate School of Medicine Dentistry and Pharmaceutical Sciences, Department of Public Health
REVIEW RETURNED	01-May-2021

GENERAL COMMENTS	The paper has been mostly revised; however, there are still several minor concerns. 1. Abstract, "Primary outcome measure" and "Results". In the revision, the authors used logistic regression models for the presence of AAC (i.e., severe and subclinical AAC) as a primary analysis. I suggest the descriptions here should be changed accordingly. 2. Table 2. I suggest the authors describe the definitions of "severe AAC" and "subclinical AAC" in the footnote of the table.
---

VERSION 2 – AUTHOR RESPONSE

Reviewer:

Comments to the Author:

The paper has been mostly revised; however, there are still several minor concerns.

Response: Thank the reviewers for their patient and professional comments. We have carefully checked and revised this manuscript to improve the readability.

1. Abstract, "Primary outcome measure" and "Results". In the revision, the authors used logistic regression models for the presence of AAC (i.e., severe and subclinical AAC) as a primary analysis. I suggest the descriptions here should be changed accordingly.

Response: Thank you for patient reading and helpful suggestions. We revised the method description in the abstract.

Odds ratio (OR) was estimated using logistic regression method to assess the association between age and the presence of severe or subclinical AAC stratified by smoking exposure (P2).

2. Table 2. I suggest the authors describe the definitions of "severe AAC" and "subclinical AAC" in the footnote of the table.

Response: We have supplemented the description of the definitions of "severe AAC" and "subclinical AAC" in the footnote of the table 2.

The 24-point semi-quantitative score of abdominal aortic calcification (AAC-24, Kauppila score) was calculated according to the length of calcification at the posterior and anterior aortic walls contiguous to the L1-L4 lumbar vertebrae. Treating lines across the middle of the intervertebral spaces as segment boundaries, the abdominal aorta was divided into eight segments. AAC was scored from 0 to 3 according to calcification length in the aortic wall of each segment (0 point: no calcification; 1 point: $\leq 1/3$ arterial wall in each segment; 2 points: $1/3 - 2/3$; 3 points: $> 2/3$). Severe AAC was defined as AAC-24 scores ≥ 6 points and subclinical AAC was defined as AAC-24 scores ≥ 2 points (Table2, P27)